# Evidence for Quasi-High-LET Biological Effects in Clinical Proton Beams That Suppress c-NHEJ and Enhance HR and Alt-EJ

**DOI:** 10.3390/cells15010086

**Published:** 2026-01-04

**Authors:** Emil Mladenov, Mina Pressler, Veronika Mladenova, Aashish Soni, Fanghua Li, Feline Heinzelmann, Johannes Niklas Esser, Razan Hessenow, Eleni Gkika, Verena Jendrossek, Beate Timmermann, Martin Stuschke, George Iliakis

**Affiliations:** 1Department of Radiation Therapy, Division of Experimental Radiation Biology, University Hospital Essen, University of Duisburg-Essen, 45147 Essen, Germany; 2Institute of Medical Radiation Biology, University Hospital Essen, University of Duisburg-Essen, 45147 Essen, Germany; 3Westdeutsches Protonentherapiezentrum Essen, University Hospital Essen, 45147 Essen, Germany; 4Institute of Cell Biology (Cancer Research), University Hospital Essen, University of Duisburg-Essen, 45147 Essen, Germany; 5Radiation Biology Laboratory, Department of Radiotherapy and Radiation Oncology, University Hospital Bonn, 53127 Bonn, Germany; 6German Cancer Consortium (DKTK), Partner Site University Hospital Essen, 45147 Essen, Germany; 7German Cancer Research Center (DKFZ), University Hospital Essen, 45147 Essen, Germany

**Keywords:** ionizing radiation (IR), linear energy transfer (LET), relative biological effectiveness (RBE), proton radiation, homologous recombination (HR), classical non-homologous end joining (c-NHEJ), alternative end joining (alt-EJ), DNA double strand breaks (DSBs), structural chromosomal abnormalities (SCAs)

## Abstract

Protons are conventionally regarded as a low-linear energy transfer (low-LET) radiation modality with a relative biological effectiveness (RBE) of 1.1, suggesting direct mechanistic similarity to X-rays in the underpinning biological effects. However, exposure to spread-out Bragg peak (SOBP) protons reveals instructive deviations from this assumption. Indeed, proton beams have a maximum LET of ~5 keV/µm but display reduced reliance on classical non-homologous end joining (c-NHEJ) as well as an increased dependence on homologous recombination (HR) and alternative end joining (alt-EJ). These features are well described in cells exposed to high-LET radiation and typically manifest between 100 and 150 keV/µm. We hypothesized that this apparent discrepancy reflects biological consequences of proton-beam properties that remain uncharacterized. In the present study, we outline exploratory experiments aiming at uncovering such mechanisms. We begin by investigating for both entrance and SOBP protons the dose-dependent engagement of HR we recently showed for X-rays. Consistent with our previous findings with X-rays, HR engagement after exposure to both types of proton beams declined with dose, from ~80% at 0.2 Gy to less than 20% at higher doses. RAD51/γH2AX foci ratios, reflecting HR engagement, were modestly higher following proton irradiation, in line with increased HR utilization. G_2_-checkpoint activation, previously linked to HR, was also stronger after exposure to protons, as was DNA end resection. Moreover, the formation of structural chromosomal abnormalities (SCAs) was higher for SOBP than entrance protons and X-rays. Collectively, our results suggest quasi-high-LET characteristics for proton beams and raise the question as to the physical proton properties that underpin them. We discuss that the commonly employed definition of LET may be insufficient for this purpose.

## 1. Introduction

Proton therapy offers highly conformal depth–dose distributions compared to photon therapy, enabling greater sparing of normal tissues and potential escalation of tumor doses [1,2,3,4,5,6,7]. These features can improve therapeutic outcomes and are particularly beneficial for pediatric tumors, where sparing adjacent developing tissues is critical, as well as for tumors with locoregional characteristics necessitating healthy tissue preservation [1,8,9]. Consequently, radiation therapy centers offering proton or other charged particle therapies are rapidly proliferating worldwide.

This expansion has driven a strong demand for a comprehensive characterization of the radiobiological properties of protons. Radiation biology classifies protons as low-linear energy transfer (LET) ionizing radiation (IR) modality, with a commonly accepted relative biological effectiveness (RBE) of approximately 1.1, implying similarity to X-rays in the underlying molecular mechanisms of action [8,10,11,12]. However, this RBE value represents an average derived from a limited number of in vitro and in vivo studies conducted under varied conditions, ignoring LET heterogeneity along the SOBP and examining only a few endpoints and cell types.

Recent findings increasingly question the validity of this assumption [13,14,15,16], arguing that it oversimplifies a more complex biological reality that introduces uncertainties in the description of the associated biological responses and may cause errors in the interpretation of adverse side-effects [17,18,19]. This outline highlights persistent gaps in our understanding of the biological effects of clinical proton beams. Addressing these gaps is essential for optimizing therapeutic outcomes, for guiding the use of DNA repair inhibitors and other DNA damage response (DDR) modifiers, and for developing clinically relevant RBE models for both normal and tumor tissues.

A central knowledge gap concerns the dynamics of proton-induced DNA double-strand break (DSB) repair. Analyses indicate deviations in the engagement of DSB repair pathways following proton versus X-ray exposure, suggesting differences in DSB complexity and subsequent pathways choice decisions [2,3,20]. In the cells of higher eukaryotes, DSBs are repaired through four pathways: classical non-homologous end joining (c-NHEJ), homologous recombination (HR), alternative end joining (alt-EJ), and single-strand annealing (SSA) [21,22,23].

Seminal studies by Pruschy showed that survival after exposure to spread-out Bragg peak (SOBP) proton beams with a maximum LET of ~10 keV/µm show no reliance on c-NHEJ and a stronger dependence on HR, compared to photon irradiation [24,25,26,27]. This is similar to what it has been reported for high-LET radiation modalities that typically manifest between 100 and 150 keV/µm and implicates high-LET effects in proton radiation response [28,29]. Indeed, reports suggest the induction of quasi-high-LET lesions after exposure to SOBP protons [24,30]. We hypothesized that this apparent discrepancy reflects physical properties of proton beams causing quasi-high-LET effects that remain uncharacterized.

The molecular mechanisms driving this divergence in response remain incompletely understood, but are likely related to shifts in DSB quality that generate shifts in the balance between c-NHEJ and DNA end resection-dependent DSB repair pathways: HR, alt-EJ, or SSA [31]. The present study aims to further characterize sources of such divergence in the DSB repair pathway balance. It was inspired by previous results from our laboratory showing that in cells irradiated with X-rays, HR engagement—measured by RAD51 recruitment—declines from over 50% at 0.25 Gy to less than 10% at 2 Gy and becomes undetectable at doses above 10 Gy [31]. This “HR confinement-effect” with increasing dose may be different with protons and thus cause changes in HR engagement.

In addition, SOBP protons, owing to their mode of generation, have mixed beam characteristics with Bragg peaks “spiked” along the plateau (five for the SOBP used here). Since most reported experiments of repair pathway engagement use SOBP protons, it is not possible to separate in the observed effects, contributions of entrance protons from those of the Bragg peak protons. To close this gap, we study here HR engagement, separately for entrance and SOBP protons and compare the results to those obtained with X-rays.

We report that the HR confinement observed with increasing photon dose is similarly observed after proton irradiation. RAD51/γH2AX foci ratios were modestly higher following proton irradiation, in line with increased HR utilization. Complementary assays related to DSB processing revealed increased DNA end resection and enhanced G_2_-checkpoint activation, also in line with increased HR engagement. Moreover, structural chromosomal abnormalities (SCAs) were slightly higher after proton exposure. Mechanistically, we propose that a quasi-high-LET component of protons suppresses c-NHEJ and favors HR, as well as possibly also other resection-dependent pathways.

## 2. Materials and Methods

### 2.1. Cell Lines, Growth and Irradiation Conditions

The A549 cell line included in the current study was obtained from the ATTC (CCL-185). The hamster cell lines AA8 and Irs1SF were provided by Dr. Larry Thompson [32], while the XR-C1-3 cells were obtained from Dr. Malgorzata Zdzienicka [33]. All cell lines were grown at 37 °C in a humidified atmosphere of 5% CO_2_ in air. A549 and hamster cell lines (DSB repair proficient-AA8, c-NHEJ deficient-XR-C1-3, and HR-deficient-Irs1SF,) were grown in McCoy’s 5A medium, supplemented with 10% fetal bovine serum (FBS). Cell lines used in the study were routinely tested for mycoplasma contamination.

Cells were exposed to X-rays at room temperature (RT) using a 320 kV, 10 mA, X-ray machine with a 1.65 mm Al filter (GE Healthcare, Freiburg im Breisgau, Germany). The dose rate at 500 mm distance from the source was 3.5 Gy/min.

Proton irradiations were performed with an IBA Proteus PLUS proton therapy treatment system (IBA PT) at the West German Proton Therapy Center, Essen (WPE). The system delivers beam energies between 100 and 230 MeV. The facility’s rooms where the experiments were conducted include a clinical pencil-beam scanning line with an IBA-dedicated nozzle and a 360°-rotatable gantry. A scanned proton field generating SOBP with a lateral field size of 20 × 20 cm^2^ and a modulation width of 1 cm was delivered to the target volume. The irradiation field was optimized with the treatment planning system RayStation^®^ version 12A (RaySearch Laboratories^®^, Stockholm, Sweden) to generate a homogeneous and flat dose distribution covering the complete volume inside the safety margin enclosing the well plates. The distribution of LET values in the SOBP of the planned field was calculated with RayStation^®^ (v12A SP1) to obtain an overview of the range of LET values inside the region of interest for the experiments. Along the depth of the plateau of the SOBP, the calculated LET values increase from 3.5 up to 6.1 keV/µm with a mean LET of 4.6 keV/µm and a median of 4.4 keV/µm inside the aforementioned safety margin. Considering the machine-dependent variation in proton beam range (+/−1 mm) and fluctuating filling levels of the cell media between different experiments, the bandwidth of calculated proton LET values inside the cell samples was calculated to be 4.1 keV/µm.

### 2.2. Indirect Immunofluorescence (IF) Analysis of γH2AX, RAD51, and pRPA32-T21 Foci

For indirect IF analysis, cells were grown on coverslips and 30 min before irradiation were incubated in media containing 2 μM 5-ethynyl-2′-deoxyuridin (EdU). Immediately thereafter, EdU was washed out and cells were irradiated with the indicated IR doses. For γH2AX or RAD51 foci analysis, cells were directly fixed for 15 min at RT in PFA-solution (3% paraformaldehyde, 2% sucrose in phosphate-buffered saline (PBS)), while the pRPA32-T21-stained cells were first pre-extracted in 0.25% Triton-X100/1xPBS for 5 min on ice and then fixed with PFA-solution. After PBS washing, cells were permeabilized for 10 min in P-Solution (50 mM EDTA, pH 8.0, 50 mM Tris-HCl, pH 7.6, 0.5% Triton-X100) and were blocked overnight at 4 °C in PBG blocking buffer (0.2% fish-skin gelatin, 0.5% BSA fraction V, in PBS).

pRPA32-T21 antibody (Abcam PLC, Cambridge, UK, ab61065) was diluted 1:400 in PBG, while the γH2AX (rabbit-polyclonal, GeneTex, Freising, Germany, GTX127342) and RAD51 (mouse-monoclonal, GeneTex, Freising, Germany, GTX70230) antibodies were diluted 1:300 and 1:400. Cells were incubated for 1.5 h at RT. The coverslips were washed three times with PBS and cells were incubated for 1.5 h at RT with anti-rabbit Alexa Fluor 555 and anti-mouse Alexa Fluor 647-conjugated secondary antibodies, diluted 1:400. The EdU signal was developed using an EdU staining kit (Click-It), (Thermo Scientific, Eindhoven, The Netherlands), according to the manufacturer’s instructions. Cells were counterstained with 0.2 µg/mL 4′,6-diamidin-2-phenylindol (DAPI) (Thermo Scientific, Eindhoven, The Netherlands) for 10 min at RT and coverslips were mounted in Immu-Mount antifade mounting media (Epredia, Runcorn, UK).

### 2.3. Quantitative Image-Based Cytometry (QIBC) Analysis in A549 Cells

AxioScan.Z1 (Carl Zeiss, Oberkochen, Germany) was utilized to scan selected areas of 4 mm × 4 mm, containing between 10.000 and 30.000 cells. QIBC analysis combining EdU and DAPI signals allowed to discriminate the cell cycle phase in which cells were at the time of irradiation. In order to follow the γH2AX, RAD51, and pRPA32-T21 foci, in selected G_2_-phase cells, cellular segmentation analyses were carried out by Imaris 9.5.1 software (Bitplane, Zürich, Switzerland) and the generated data was converted to a proper format for analysis in Kaluza 2.1 software (Beckman Coulter, Krefeld, Germany), where the specific gates were applied (Appendix A).

### 2.4. Two-Parametric Flow Cytometry Analysis to Evaluate Activation and Recovery of the G_2_-Checkpoint in G_2_-Irradiated Cells

Cells were harvested at various times after IR and fixed at −20 °C in 70% ethanol. After fixation, cells were resuspended in 1 mL of 0.25% Triton-X100 in PBS and processed as previously described [34]. Samples were acquired by Gallios flow cytometer (Beckman Coulter, Krefeld, Germany) and analyzed using Kaluza 2.1 software. To calculate the mitotic index (MI), the number of H3-pS10-positive cells and the total number of cells were assessed. Results are shown as normalized MI as a function of the post-irradiation incubation time [34].

### 2.5. Multicolor Fluorescence “In Situ” Hybridization (mFISH) Analysis of Hamster Cell Lines

The mFISH analysis is performed as described previously [29]. Briefly, to accumulate cells at metaphase, colcemid (Biochrom AG, Berlin, Germany) was added for 2–3 h at a concentration of 0.1 μg/mL. Metaphase spreads were prepared using standard cytogenetic procedures. mFISH was performed using 12XCHamster Multicolor FISH Probe for Chinese Hamster Chromosomes (MetaSystems, Altlußheim, Germany) according to manufacturer’s protocol. An automated imaging system (MetaSystems, Altlußheim, Germany) was used to obtain high-quality images of metaphase chromosomes that were further processed by the Isis software v6.2 (MetaSystems, Altlußheim, Germany). For analysis, at least 100 metaphases were scored in each of at least two independent experiments. The number of recurrent SCAs, determined by generating the karyotype of AA8 cells, was subtracted from the number of radiation-induced SCAs.

### 2.6. Statistical Analysis

Statistical analyses were performed by using the comparison of means calculator developed by MedCalc (https://www.medcalc.org/calc/comparison_of_means.php, Version 23.3.7; accessed 30 August 2025).

## 3. Results

### 3.1. Increased Dependence on HR Rather than c-NHEJ for Survival in Cells Exposed to Protons Versus X-Rays

In repair-proficient AA8 cells, entrance and SOBP protons show killing efficacy broadly similar to that of X-rays (Figure 1A and Appendix A). Upon close inspection, entrance protons appear less effective than SOBP protons, but the difference does not reach statistical significance. Thus, putative differences in DNA damage quality induced by the two types of proton beams are of little consequence to cell killing when all DSB repair pathways are active—likely because of functional complementation.

Suppression of c-NHEJ by inactivation of DNA-PKcs in the XR-C1-3 CHO mutant cells dramatically increases radiosensitivity, but strikingly, the results obtained after exposure of this cell line to X-rays, entrance, or SOBP protons are indistinguishable (Figure 1B and Appendix A). Thus, putative changes in DSB quality, associated with exposure to entrance or SOBP protons, fail to generate changes in radiosensitivity when c-NHEJ is inactive. This is reminiscent of high-LET radiation responses, where c-NHEJ-deficient mutants fail to show enhanced killing with increasing LET, and they indeed display an RBE of approximately 1 after exposure to α-particles or heavy ions [35,36,37,38,39,40]. This suggests that with increasing LET a shift occurs in the quality of the DSB-subset that is responsible for cell killing (~10% of total), such that c-NHEJ fails to engage. Thus, high-LET radiation modalities can be regarded as potent c-NHEJ suppressors with effectiveness approaching that of DNA-PKcs inactivation. However, proton beams have a maximum LET of ~5 keV/µm, while high-LET radiation effects typically manifest between 100 and 150 keV/µm. We therefore infer quasi-high-LET effects after proton exposure that are not reflected by the numerical values of their LET.

High-LET-mediated suppression of c-NHEJ is expected to unleash resection-dependent pathways, such as HR and alt-EJ, to remove DSBs. It can therefore be expected that defects in HR will radiosensitize cells, with greater effect after exposure to SOBP protons, owing to their increased quasi-high-LET component. In full agreement with this expectation, markedly increased radiosensitivity is observed with XRCC3 deficient, Irs1SF cells after exposure to SOBP protons, as compared to X-rays (Figure 1C and Appendix A). However, entrance protons also show increased radiosensitization, suggesting that characteristics beyond high-LET spiking through Bragg peaks underpin the biological responses of proton beams. Appendix A shows published results [27] obtained with the same mutants and plotted as in Figure 1 to facilitate comparison; it is evident that our results are in broad agreement with this early report [27].

### 3.2. Enhanced Recruitment of RAD51 Protein to DSBs After Exposure to SOBP Protons Versus X-Rays

The increased contribution of HR to cell survival following proton irradiation raises the question of whether enhanced HR utilization is also detectable at the level of individual DSBs—i.e., through increased recruitment of RAD51 protein to DNA damage-induced foci. Despite the relevance of this question in protons versus X-ray studies, limited number of reports analyzing RAD51 foci are available [15,41]. Studies of HR engagement by means of RAD51 foci analysis are particularly relevant to our work for an additional reason. We have recently shown using RAD51 foci analysis that HR is strongly suppressed with increasing X-ray doses [31]. It was therefore important to investigate whether and to what extent similar confinement effects are observed with increasing proton doses.

We used immunofluorescence (IF) to detect γH2AX and RAD51 foci in A549 cells, specifically selecting cells in the G_2_-phase of the cell cycle, where HR activity is at its maximum [31] (Figure 2).

To distinguish G_2_-phase cells from those in S-phase, we labeled the S-phase population with EdU, 30 min. prior to irradiation. In the subsequent analysis, we excluded the fraction of EdU-positive (EdU^+^) cells that progress from S-phase to G_2_-phase after irradiation (see Section 2). As shown in Figure 2 and Appendix A, cells exposed to increasing doses of both protons and X-rays develop distinct γH2AX foci in a dose-dependent manner. A parallel increase in RAD51 foci formation is also observed, indicating HR engagement at DSBs.

The gating strategy used to separate EdU-negative, G_2_-phase cells (EdU^−^, G_2_-cells) is shown in Appendix A, while Appendix A presents the distribution of γH2AX foci in the gated population and illustrates that a large number of cells are processed. Quantification of γH2AX foci at 1 h post-irradiation reveals a linear dose–response (Figure 2A), with no relevant statistically significant differences among the radiation types tested. In contrast to γH2AX foci, which are analyzed at 1 h, where they uniformly reach their peak, detailed RAD51 foci kinetics are generated, with the maximum foci count, reached at increasing times with increasing dose. The number of RAD51 at maximum is used to construct the dose–response curves, as previously described [31]. As shown in Figure 2B, the RAD51 dose–response curve for X-rays shows the expected pattern [31], with HR engagement plateauing above 1 Gy. Entrance protons yield a similar plateau, whereas SOBP protons show significantly higher RAD51 foci levels in the plateau, suggesting increased engagement of HR.

To evaluate HR activity relative to the overall DNA damage load (number of DSBs induced), we calculated the ratio of RAD51 to γH2AX foci at each radiation dose [31]. Figure 2C reproduces the previously reported decrease in HR engagement with increasing X-ray dose. A quantitatively similar trend is observed for entrance protons. In contrast, SOBP protons induce a generally higher level of HR engagement across all doses.

Since enhanced RAD51 recruitment suggests active DNA end resection [42], we also quantified in parallel experiments pRPA32-T21 foci—an established marker of single-stranded DNA at resected DSB ends. As shown in Figure 2D, cells exposed to SOBP protons exhibit greater DNA end resection than those treated with entrance protons or X-rays, in line with the increase in HR engagement noted above. However, DNA end resection also supports alt-EJ [20,43]. We return to this point below when we analyze formation of chromosomal alterations in irradiated cells.

### 3.3. Stronger G_2_-Checkpoint Activation After Exposure to Protons

We have previously reported a link between HR activity and G_2_-checkpoint activation in cells specifically irradiated during the G_2_-phase of the cell cycle. Indeed, HR-deficient mutants are severely impaired in initiating and maintaining the G_2_-checkpoint at low radiation doses, making checkpoint strength an indirect measure of HR activity [34]. Accordingly, we expected a stronger checkpoint response in cells exposed to protons than to X-rays owing to the increased engagement of HR.

To test this postulate, we used two-parametric flow cytometry to measure the mitotic index (MI), calculated as the fraction of cells positive for the M-phase-specific marker, histone H3-pS10, over time following exposure to 1 Gy (Appendix A). The sharp decline in mitotic cells observed 1 h after X-ray irradiation (Figure 3A) demonstrates activation of the G_2_ checkpoint, as the arrest of cells in G_2_-phase depletes the M-phase compartment. Recovery in the fraction of cells in M-phase begins after 1 h and is complete by 5 h, indicating checkpoint release.

Notably, proton irradiation leads to markedly stronger checkpoint activation, with similar effects observed for both entrance and SOBP protons. The checkpoint remains active here for 1–3 h post-irradiation, after which recovery begins. Thus, protons induce a markedly stronger and more prolonged G_2_-checkpoint response compared to X-rays. This outcome aligns with expectations of increased HR engagement.

At this low dose, and as previously reported, checkpoint activation is epistatically regulated by ATM and ATR [44]. Consistently, inhibition of either kinase with specific inhibitors fully abrogates the checkpoint response (Figure 3B,C). We conclude that the enhanced checkpoint activation observed after proton exposures reflects augmented HR engagement in G_2_-phase cells, subject to similar regulation by ATM and ATR as in cells exposed to X-rays.

### 3.4. Enhanced Formation of Chromosomal Abnormalities After Exposure to Protons

In further experiments, we analyzed the effect of radiation modality on the formation of SCAs in AA8 cells using mFISH analysis (Figure 4A). As indicated above, increase in resection also favors alt-EJ that is directly linked to this endpoint. For karyotyping the AA8 cell line, we followed the previously published chromosome annotation of CHO cells [29,45,46]. The karyotype of AA8 cells shows recurrent chromosomal abnormalities compared to the CHO K1 cell line, which is the origin of this cell line (Figure 4A).

AA8 cells were exposed to 1 Gy X-rays, entrance, or SOBP protons and harvested for mFISH analysis 24 h later (Figure 4B). Simple and complex SCAs were analyzed. Examples of aberrations are shown in Figure 4B and the results in Figure 4C,D. It is evident that after exposure to SOBP protons, a statistically significant increase in both simple and complex chromosomal aberrations is observed (Figure 4B–D). These results confirm that increased DNA end resection after proton irradiation promotes alt-EJ.

## 4. Discussion

### 4.1. DSBs Induced by Protons Show Dose-Dependent HR Confinement Similar to X-Rays

The work summarized above adds to a body of evidence suggesting increased relative engagement of HR and alt-EJ in the repair of proton-induced DSBs. While previous work was mostly carried out with SOBP protons, we show here qualitatively similar responses with cells exposed to entrance protons. Our results extend previous observations on increased radiosensitivity of HR-deficient cells to protons versus X-rays and also demonstrate increased formation of RAD51 foci in proton-exposed cells.

Extensive work from our laboratory has recently demonstrated a strong suppression of HR with increasing X-ray dose, an effect delimited but not determined by either 53BP1 or RAD52 [22]. The effect is considerable with over half of the generated DSBs processed by HR at lower IR doses, while this contribution becomes negligible at doses above 10 Gy. Notably, radiation dose-dependent suppression of HR engagement is also observed with increasing proton dose with a dose–response very similar to that observed in parallel experiments carried out with the same pool of cells after exposure to X-rays.

Thus, the determinants of HR suppression after exposure to protons likely have similar mechanistic underpinnings to X-ray exposure. This is important in the clinical application of protons, particularly when selected as treatment modality on the basis of their HR preference, i.e., in cancers characterized by “BRCAness”. The dose-dependent reduction in HR suggests that the expected differential effect will be considerably stronger at low rather than high radiation doses.

It is worth noting that at all examined endpoints, the effect generated with entrance protons was lower than with SOBP protons, although the difference often failed to reach statistical significance. Notably, the increased engagement of HR after exposure to both types of protons could be convincingly demonstrated by their effects at the checkpoint control level. The reasons for the sensitivity of this endpoint to proton characteristics remain unknown. However, the strong response may prove useful in the development of predictive assays.

### 4.2. A Quasi High-LET Radiation Component in Proton Beams May Underpin Shifts to HR and Alt-EJ

While several reports, including the present one, are in line with increased utilization of HR after exposure to protons than X-rays, the underpinning mechanisms remain elusive [15,24,25,26,27,47,48,49]. What causes the increased HR engagement in proton-exposed cells? As outlined in the Introduction, we postulate that this increase stems from changes in the quality of the underpinning DNA lesions, i.e., the quality of the DSBs, that alter the balance of repair pathway engagement.

The exact form of DSB quality change remains unknown, but it phenomenologically bears similarities to responses noted after exposure to high-LET radiation modalities. Indeed, high-LET radiation modalities change the quality of induced DSBs in ways that cause a collapse in the function of c-NHEJ and thus give way to resection-dependent pathways, HR and alt-EJ. If such a mechanism is active after irradiation with protons, it would explain the increased engagement of HR.

The apparent problem with this interpretation is that protons are low-LET radiation modality, with LET values even at the Bragg peak being less than ~5 keV/µm. How could they be generating effects that are characteristic of high-LET radiation typically manifesting at values between 100 and 150 keV/µm? Our working hypothesis is that properties of protons that cannot be described by the LET alone underpin the observed effects.

While devising experiments to characterize these DSBs, it is important to keep in mind that the suppression of c-NHEJ in cells exposed to high LET IR is confined to those lesions that determine cell killing, i.e., a specific small subset of DSBs (~10%). Indeed, when the entirety of DSBs is analyzed for repair pathway engagement, a large proportion (80–90%) is processed by c-NHEJ [21,35]. Our working hypothesis is that the biologically relevant form of DSB complexity is DSB clusters and we have extensively published on the biological consequences of such lesions [21,28,29,50,51].

Summary: In the present work, we compare results obtained with protons and X-rays at the DNA, cellular, and ultimately the chromosome levels. We demonstrate that HR and alt-EJ are preferentially engaged in the repair of proton-induced DSBs. This preference reflects enhanced DNA end resection deriving from the inhibition of c-NHEJ, at a specific subset of proton-induced DSBs. These insights will help develop experimental strategies to better characterize the biological effects of proton irradiation, including character of the relevant DSBs, and optimize their clinical application.

## Figures and Tables

**Figure 1 cells-15-00086-f001:**
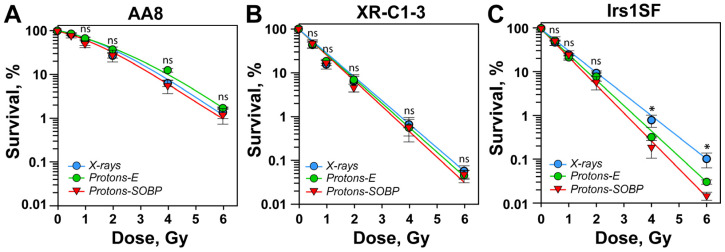
Homologous recombination deficient cells are more radiosensitive when irradiated with SOBP protons. (**A**) Clonogenic survival data of DSB repair-proficient AA8 hamster cells exposed to increasing doses of X-rays, entrance protons (Protons-E), and protons generated at the SOBP (Protons-SOBP). (**B**) Same as in panel A, but for XR-C1-3 cells that are c-NHEJ-deficient. (**C**) Same as panel A, but for HR-deficient, Irs1SF cells. All experiments represent the results from at least three biological repeats, with the mean and standard deviations. The following annotations were used to indicate the significance level: ns (*p* > 0.05), * (*p* ≤ 0.05).

**Figure 2 cells-15-00086-f002:**
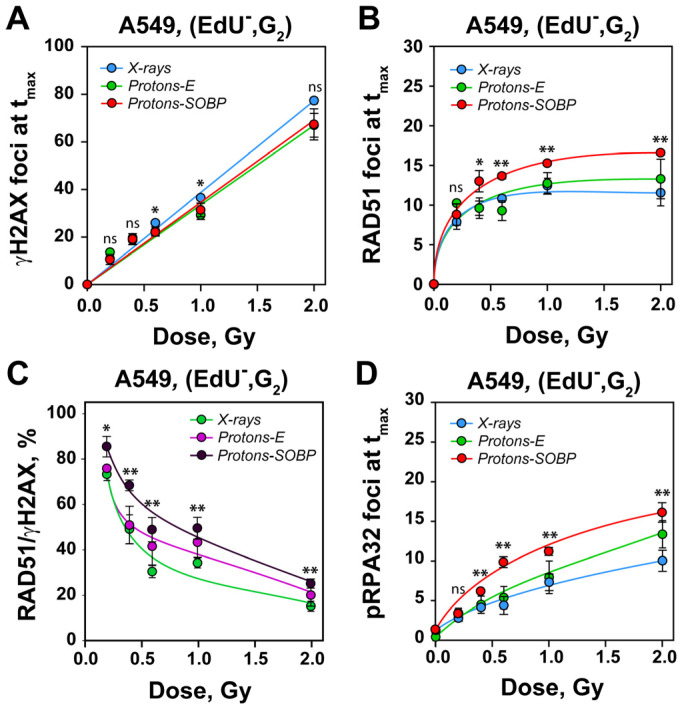
Irradiation of A549 cells with X-rays and protons at the entrance or at SOBP revealed linear, dose-dependent increase in γH2AX foci and non-linear dose response for RAD51 and pRPA32-T21 foci in G_2_-phase irradiated cells. (**A**) Dose-dependent increase in γH2AX foci in A549 cells irradiated in the G_2_-phase of the cell cycle (EdU^−^, G_2_) with increasing doses of X-rays, protons at the entrance (Protons-E), and the SOBP (Protons-SOBP). (**B**) Dose–response curves of RAD51 foci in EdU^−^, G_2_-phase, A549 cells irradiated with increasing doses of X-rays, protons at the entrance (Protons-E), and the SOBP (Protons-SOBP). (**C**) Ratio of RAD51 and γH2AX foci in A549 cells exposed to selected IR modalities. The RAD51/γH2AX ratio represents the fraction of IR-induced DSBs that are repaired by HR and is calculated by taking the number of γH2AX foci at their maximum (t_max_), 1 h post-IR, and the numbers of RAD51 foci at t_max_, which for the applied doses was between 1 and 3 h post-irradiation. (**D**) Dose-dependent increase in pRPA32-T21 (pRPA32) foci in EdU^−^, G_2_-phase, and A549 cells irradiated with increasing doses of selected radiation modalities. All data represent the mean and SD from three independent determinations. The following annotations were used to indicate the significance level: ns (*p* > 0.05), * (*p* ≤ 0.05), ** (*p* ≤ 0.01).

**Figure 3 cells-15-00086-f003:**
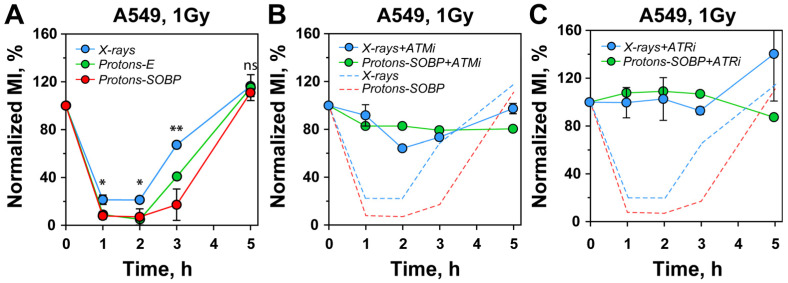
Exposure of A549 cells to protons results in increased dose-dependent activation of the G_2_-checkpoint for cells irradiated in G_2_-phase of the cell cycle. (**A**) Normalized MI of A549 cell irradiated with 1 Gy of X-rays, protons at the entrance (Protons-E), and the SOBP (Protons-SOBP). (**B**) Normalized MI of A549 cell irradiated with 1 Gy as in A in the presence of ATM inhibitor, KU55399 (ATMi). The dashed lines represents the data for X-rays and SOBP protons, plotted in panel A. (**C**) Normalized MI of A549 cell irradiated with 1 Gy as in A in the presence of ATR inhibitor, VE-821 (ATRi). The dashed lines represents the data for X-rays and SOBP protons, plotted in panel A. All data represents the mean and SD from three independent biological repeats. The following annotations were used to indicate the significance level: ns (*p* > 0.05), * (*p* ≤ 0.05), ** (*p* ≤ 0.01).

**Figure 4 cells-15-00086-f004:**
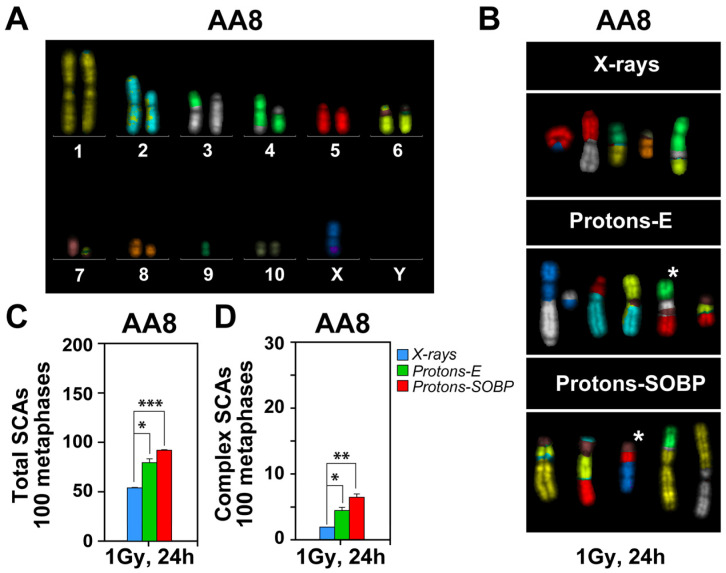
Increased formation of simple and complex SCAs in AA8 cells exposed to SOBP protons. (**A**) Karyotype of AA8 cells, generated by mFISH analysis, showing recurrent numerical and structural abnormalities in the identified hamster chromosomes (1-10, X, and Y). (**B**) Representative SCAs obtained after mFISH analysis in AA8 cells, exposed to protons and X-rays. Complex SCAs are marked with asterisk (*).Quantification of total (**C**) and complex (**D**) SCAs in parental AA8 cells, 24 h after exposure to 1 Gy, as indicated. Data represent the mean and SD from at least two biological repeats. The following annotations were used to indicate the significance level: * (*p* ≤ 0.05), ** (*p* ≤ 0.01), *** (*p* ≤ 0.001).

## Data Availability

The raw results and the data generated through the current study is available upon request from the corresponding authors.

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
