# Peer review of "Evidence for Quasi-High-LET Biological Effects in Clinical Proton Beams That Suppress c-NHEJ and Enhance HR and Alt-EJ"

_cells, 2026, doi:10.3390/cells15010086_

Round 1

Reviewer 1 Report

Comments and Suggestions for Authors

This is an important study comparing DNA repair mechanisms related to c-NHEJ and HR for proton irradiation including SOBP (1 cm width) to X-rays. This study provides important information in Figures 2 and 3, while Figure 4 on cytogenetic results strengthens the study.

A suggestion is to provide more information on the proton irradiation. The method use in the plateau is for several energies (needed to make the SOBP) so the average LET would be useful (~0.4 keV/micron). Then a SOBP is used of 1 cm width so a mixture of LETs with a guess of an average of 10 keV/micron. Can the physicists at the facility provide the average LET?

Also the X-rays used are 90 kVp. This is a bit low with 200 to 250 kVp more often used. This probably increased the X-ray effectiveness and narrowed the differences with the proton results.

Author Response

Reviewer 1

[Regarding major comments]

“This is an important study comparing DNA repair mechanisms related to c-NHEJ and HR for proton irradiation including SOBP (1 cm width) to X-rays. This study provides important information in Figures 2 and 3, while Figure 4 on cytogenetic results strengthens the study.”

We really appreciate the positive evaluation and the recognition of the significance and importance of our study.

“A suggestion is to provide more information on the proton irradiation. The method use in the plateau is for several energies (needed to make the SOBP) so the average LET would be useful (~0.4 keV/micron). Then a SOBP is used of 1 cm width so a mixture of LETs with a guess of an average of 10 keV/micron. Can the physicists at the facility provide the average LET?”

The suggested information is now added and the corresponding Materials and Methods paragraph is extended according to the reviewer’s suggestion.

“Also the X-rays used are 90 kVp. This is a bit low with 200 to 250 kVp more often used. This probably increased the X-ray effectiveness and narrowed the differences with the proton results.”

We thank the reviewer for pointing out this potentially misleading piece of information. We clarify that in our study we used a 320 kV, 10 mA, X-ray machine with a 1.65 Al filter. The information referred by the reviewer was about the mean electron energy generated in water from photons from such a beam. To avoid confusion, and because this is not essential for the understanding of the results provided, we omitted this information in the revised version of the manuscript.

Reviewer 2 Report

Comments and Suggestions for Authors

The manuscript “Augmented homologous recombination activity in proton-exposed cells boosts checkpoint response but is suppressed with increasing radiation dose” reports the results of the investigation that aims to clarify the role of two main DNA double-strand break (DSB) repair mechanisms, namely non-homologous end joining (NHEJ) and homologous recombination (HR),  in the response of cells to ionising radiation of different qualities. Undoubtfully, this subject is important for both fundamental and applied aspects of radiobiology. Understanding of mechanisms of DNA DSB repair following exposure to entrance and spread-out Bragg peak (SOBP) protons might be useful for optimisation of proton beam cancer radiotherapy. Therefore, the subject of the study presents an interest for a wide audience of researchers including experimental radiobiologists and clinical radiation oncologists. The work exploits a broad range of modern experimental methods. The manuscript is logically well structured and written in appropriate scientific language, nevertheless, there is a room for further improvement of the paper by addressing a few issues detailed below in General and Specific comments.

General comments

  1. Two modes of proton beam are investigated in the study – entrance and SOBP protons. The information on physical beam properties (beam energy, linear energy transfer – LET) are not provided in the manuscript. While for entrance protons these properties are more or less known, SOPB properties might vary depending on particular installation, and the contribution of so-called “high-LET” component is important. Therefore, the information on dose-averaged LET for SOBP proton beam would useful for adequate interpretation of the results.
  2. One of the interesting observations reported in the study is the lack of differences in cell survival between all three types of irradiation – X-rays, entrance and SOBP protons for both wild type and NHEJ-deficient mutant (Figures 1A and B). The authors draw a parallel between NHEJ-deficiency and exposure to high-LET irradiation, for which an RBE (Relative Biological Effectiveness) is approximately 1 for NHEJ-deficient mutant.   It is stated that “This is reminiscent to high-LET radiation responses, where c-NHEJ deficient mutants fail to show enhanced killing with increasing LET”. This statement is ambiguous, since the meaning of “enhanced killing” is not clear: enhanced compared to wild type cells or enhanced compared to low LET radiation? In the present study, c-NHEJ deficient mutants shows enhanced killing compared to wild type cells with increasing LET, however fails to show enhanced killing for SOBP protons compared to X-rays.
  3. Another questionable point is an inference of the higher contribution of “high-LET” component for SOBP protons as compared to entrance protons. First, this is not obvious for the results obtained for both wild type and NHEJ deficient cells given that RBE of both protons beams are close to 1. Second, it depends on the meaning of “high LET” term. In the context of cultured mammalian cells survival, it is common to consider high LET as those values, at which peak RBE achieved, and those values are above 100 keV/micron. Dose averaged LET value for SOBP protons might vary depending on particular type of installation and depth, with peak values, for example, from 10 to 30 keV/micron and substantially smaller average values (Biomed. Phys. Eng. Express 10 (2024) 035004). These values are much smaller than “high LET” of 100 keV/micron. In this context, accurate usage of “high LET” term and addition of information on dose averaged LET for SOBP (comment 1) is recommended.
  4. Consideration of high LET radiations as “a potent NHEJ suppressors” and similar suggestion that “with increasing LET… NHEJ fails to engage” do not shed much light on the mechanism of such suppression or failure. One possible mechanism is pointed out in the manuscript that is related to the change in the spectrum of induced DNA lesions, namely increasing fraction of more complex lesions (DNA DSB) with increasing LET. Presumably, the “failure to engage” means the inability of NHEJ to repair such more complex lesions(?). There are however other mechanisms that can be manifested as “NHEJ failure to engage”. This can be the change in the distribution of cells with regard to the number of DNA lesions. Due to the increased probability with increasing LET of induction of a few lesions by a single particle, the distribution deviates from Poisson statistics with increasing LET, and the fraction of “lightly” damaged cells will decrease and the fraction of “heavily” damaged cells will increase. Then the impact of DNA repair (even if it is engaged) on the survival will be less pronounced. An ultimate case is a high LET (at and beyond RBE peak), at which one particle crossing a cell generates an excessive number of lesions (saturation effect), so that the fully engaged repair system cannot cope with such damage. The interpretation of the results in Figure 1A and B in terms of NHEJ failure to engage, however, seems to be irrelevant (or has a negligible contribution) to the reported results, since NHEJ failure to engage in the case of SOBP protons, if occurring, would result in increased radiosensitivity of wild type cells to SOBP protons, which is not observed in the present study (Figure 1A).
  5. Another non-trivial observation, reported in the study, is an apparent higher enhancement of radiosensitivity by HR deficiency in case of SOBP protons as compared to X-rays, that follows from the inspection of Figures 1 A and C. First, it would be useful to numerically estimate this HR enhancement ratio to confirm this observation. This observation means higher RBE of SOBP protons for HR deficient cells compared to wild type. Such RBE values are somewhat counterintuitive given that usually peak RBE values for radiosensitive cells (repair deficient mutants) are lower than for wild type cells (as also stated in line 187). The authors’ interpretation of this observation is that “High-LET mediated suppression of c-NHEJ is expected to unleash resection-dependent pathways, such as HR and alt-EJ, to remove DSBs”. While generally reasonable, this interpretation is not supported by the results in Figure1A, that does not indicate the suppression of NHEJ pathway (as stated in Comment 4). This observation, however, can be explained by assuming that HR pathway is able to repair more complex lesions, as compared to NHEJ. With regard to RBE values for SOBP protons, their LET value is probably much lower than LETs required for peak RBE values, which are expected to be higher for wild type than for HR mutant.
  6. The most intriguing finding of the manuscript (and previous work) is the decline of HR engagement with increasing dose. This hypothesis, while supported generally by experimental results, raises questions that are not answered and not discussed in the study. It is expected that the dose-declining repair process would result in sigmoidal survival curve (as far as this repair process deals with critical lesions). Such a curve is observed for X-rays but not for protons. Does it mean that another repair process increases its contribution with increasing dose? If so, how it agrees with the conclusion of decreasing role of NHEJ for protons as compared to X-rays? It is also reasonable to expect that complex chromosome aberrations emerge as a result of mis-joining in the process of NHEJ which is error-prone. In this context, how the increase in the number of complex aberrations for SOBP agrees with decline of NHEJ? Addressing these issues in the discussion would beneficial for the paper.   

Specific comments

  1. The statement (page 9 lines 369-370) “Indeed, the results presented in Figure 1 show that cell survival is independent of c-NHEJ function, which is equivalent to a partial collapse of this pathway in repair proficient cells” is simply wrong. The results in Figure 1 shows that survival depends on c-NHEJ function and the enhancement caused by the deficiency of this function is independent of the type of irradiation, or in other words, RBE of protons is independent of c-NHEJ function. Calculation and reporting of enhancement ratio produced by repair deficiency would clarify this issue.
  2. The statement (page 10 lines 378-380) “The experimental testing of this hypothesis is confounded by the fact that while c- NHEJ entirely fails to contribute to cell survival after exposure to high-LET radiation modalities and RBE value of 1.1 is measured, it has a marked contribution when repair of all DSBs is assessed” is ambiguous since it is not clear what cells are meant – wild type or NHEJ deficient mutant. For wild type cells, NHEJ fails to contribute to cell survival at high LET resulting in high RBE values (~3 but not 1.1); for NHEJ deficient mutant, NHEJ cannot fail at high LET since it has already failed by definition (at any LET) resulting in low RBE (~1).
  3. Lines 77-78 page 2: What is meant by “high-LET quasi lesions”? Should it be “quasi high-LET lesions”?
  4. Lines 110 and 112 page 3: the energy of X-rays is measured in keV but not in kV. The operating voltage of X-ray tube is measured in kV.
  5. Figure 4 legend – AA8 cells (wild type) label placed in Figures С and D and the reference to AA8 cells made in the text, however “HR-deficient cells” are indicated in the first sentence in the legend.

Author Response

Reviewer 2

“The manuscript “Augmented homologous recombination activity in proton-exposed cells boosts checkpoint response but is suppressed with increasing radiation dose” reports the results of the investigation that aims to clarify the role of two main DNA double-strand break (DSB) repair mechanisms, namely non-homologous end joining (NHEJ) and homologous recombination (HR),  in the response of cells to ionising radiation of different qualities. Undoubtfully, this subject is important for both fundamental and applied aspects of radiobiology. Understanding of mechanisms of DNA DSB repair following exposure to entrance and spread-out Bragg peak (SOBP) protons might be useful for optimisation of proton beam cancer radiotherapy. Therefore, the subject of the study presents an interest for a wide audience of researchers including experimental radiobiologists and clinical radiation oncologists. The work exploits a broad range of modern experimental methods. The manuscript is logically well structured and written in appropriate scientific language, nevertheless, there is a room for further improvement of the paper by addressing a few issues detailed below in General and Specific comments.”

We thank the Reviewer for his positive evaluation of our work and appreciate the provided constructive comments, which we address below:

General comments

“1.          Two modes of proton beam are investigated in the study – entrance and SOBP protons. The information on physical beam properties (beam energy, linear energy transfer – LET) are not provided in the manuscript. While for entrance protons these properties are more or less known, SOPB properties might vary depending on particular installation, and the contribution of so-called “high-LET” component is important. Therefore, the information on dose-averaged LET for SOBP proton beam would useful for adequate interpretation of the results.”

The information regarding the physical properties of the proton beams at the entrance and at the SOBP are now included in the corresponding section of the Materials and Methods.

“2.          One of the interesting observations reported in the study is the lack of differences in cell survival between all three types of irradiation – X-rays, entrance and SOBP protons for both wild type and NHEJ-deficient mutant (Figures 1A and B). The authors draw a parallel between NHEJ-deficiency and exposure to high-LET irradiation, for which an RBE (Relative Biological Effectiveness) is approximately 1 for NHEJ-deficient mutant.   It is stated that “This is reminiscent to high-LET radiation responses, where c-NHEJ deficient mutants fail to show enhanced killing with increasing LET”. This statement is ambiguous, since the meaning of “enhanced killing” is not clear: enhanced compared to wild type cells or enhanced compared to low LET radiation? In the present study, c-NHEJ deficient mutants shows enhanced killing compared to wild type cells with increasing LET, however fails to show enhanced killing for SOBP protons compared to X-rays.”

We thank the Reviewer for pointing out this ambiguity in the design of our work and the discussion of experiments. In response to this criticism (and more that follow), we have reorganized relevant sections of the paper. Specifically, we changed its title to reflect overarching effects and modified the abstract to more accurately outline our initiating model and the conclusions based on the results obtained. We also made numerous changes in the results section and edited the discussion to edit or remove erroneous or incorrect statements.

“3.          Another questionable point is an inference of the higher contribution of “high-LET” component for SOBP protons as compared to entrance protons. First, this is not obvious for the results obtained for both wild type and NHEJ deficient cells given that RBE of both protons beams are close to 1. Second, it depends on the meaning of “high LET” term. In the context of cultured mammalian cells survival, it is common to consider high LET as those values, at which peak RBE achieved, and those values are above 100 keV/micron. Dose averaged LET value for SOBP protons might vary depending on particular type of installation and depth, with peak values, for example, from 10 to 30 keV/micron and substantially smaller average values (Biomed. Phys. Eng. Express 10 (2024) 035004). These values are much smaller than “high LET” of 100 keV/micron. In this context, accurate usage of “high LET” term and addition of information on dose averaged LET for SOBP (comment 1) is recommended.”

We also agree with the criticism of the Reviewer and have rephrased relevant sections. Other relevant changes are as outlined under Comment 2.

“4.          Consideration of high LET radiations as “a potent NHEJ suppressors” and similar suggestion that “with increasing LET… NHEJ fails to engage” do not shed much light on the mechanism of such suppression or failure. One possible mechanism is pointed out in the manuscript that is related to the change in the spectrum of induced DNA lesions, namely increasing fraction of more complex lesions (DNA DSB) with increasing LET. Presumably, the “failure to engage” means the inability of NHEJ to repair such more complex lesions(?). There are however other mechanisms that can be manifested as “NHEJ failure to engage”. This can be the change in the distribution of cells with regard to the number of DNA lesions. Due to the increased probability with increasing LET of induction of a few lesions by a single particle, the distribution deviates from Poisson statistics with increasing LET, and the fraction of “lightly” damaged cells will decrease and the fraction of “heavily” damaged cells will increase. Then the impact of DNA repair (even if it is engaged) on the survival will be less pronounced. An ultimate case is a high LET (at and beyond RBE peak), at which one particle crossing a cell generates an excessive number of lesions (saturation effect), so that the fully engaged repair system cannot cope with such damage. The interpretation of the results in Figure 1A and B in terms of NHEJ failure to engage, however, seems to be irrelevant (or has a negligible contribution) to the reported results, since NHEJ failure to engage in the case of SOBP protons, if occurring, would result in increased radiosensitivity of wild type cells to SOBP protons, which is not observed in the present study (Figure 1A).”

We thank the reviewer for this very insightful outline. Inquiries like those raised by the reviewer are at the heart of our work and central to ongoing or planned projects. Since at the moment we are in the realm of speculation, we opted not to burden the uninitiated reader with related speculations. We point out however in the extensively revised manuscript the direction our thinking goes, the working model we use and the type of projects that may come next.

“5.          Another non-trivial observation, reported in the study, is an apparent higher enhancement of radiosensitivity by HR deficiency in case of SOBP protons as compared to X-rays, that follows from the inspection of Figures 1 A and C. First, it would be useful to numerically estimate this HR enhancement ratio to confirm this observation. This observation means higher RBE of SOBP protons for HR deficient cells compared to wild type. Such RBE values are somewhat counterintuitive given that usually peak RBE values for radiosensitive cells (repair deficient mutants) are lower than for wild type cells (as also stated in line 187). The authors’ interpretation of this observation is that “High-LET mediated suppression of c-NHEJ is expected to unleash resection-dependent pathways, such as HR and alt-EJ, to remove DSBs”. While generally reasonable, this interpretation is not supported by the results in Figure1A, that does not indicate the suppression of NHEJ pathway (as stated in Comment 4). This observation, however, can be explained by assuming that HR pathway is able to repair more complex lesions, as compared to NHEJ. With regard to RBE values for SOBP protons, their LET value is probably much lower than LETs required for peak RBE values, which are expected to be higher for wild type than for HR mutant.”

In the revised version of the manuscript, a supplementary table (Table S1), containing the numerical values of the survival experiments shown in Figure 1 is included. The table also includes the calculated RBE values at 50% (D50), 37% (D37), 10% (D10), and 1% (D1) survival. Again, we avoided speculation, but clearly describe our working hypothesis.

“6.          The most intriguing finding of the manuscript (and previous work) is the decline of HR engagement with increasing dose. This hypothesis, while supported generally by experimental results, raises questions that are not answered and not discussed in the study. It is expected that the dose-declining repair process would result in sigmoidal survival curve (as far as this repair process deals with critical lesions). Such a curve is observed for X-rays but not for protons. Does it mean that another repair process increases its contribution with increasing dose? If so, how it agrees with the conclusion of decreasing role of NHEJ for protons as compared to X-rays? It is also reasonable to expect that complex chromosome aberrations emerge as a result of mis-joining in the process of NHEJ which is error-prone. In this context, how the increase in the number of complex aberrations for SOBP agrees with decline of NHEJ? Addressing these issues in the discussion would beneficial for the paper.”

We thank the reviewer for appreciating the gravity of one of our major recent discoveries. In the relevant publication (Mladenov, E., et al., Strong suppression of gene conversion with increasing DNA double-strand break load delimited by 53BP1 and RAD52. Nucleic Acids Res, 2020. 48: p. 1905-1924.), we investigated in detail the observed dose dependent decline in the efficiency of HR with increasing the radiation dose and discussed its potential relevance to the shoulder of the survival curve. We do not know yet what causes this decline, but we actively explore mechanisms. We hypothesize that the underpinning processes after exposure to protons are the same as after X-rays and discuss this aspect in the revised manuscript. We also emphasize that c-NHEJ suppression with increasing LET is only relevant for a ~10% subset of DSBs. When the bulk of DSBs is examined, for example using PFGE, large contributions of c-NHEJ are measured.

Specific comments

“1.          The statement (page 9 lines 369-370) “Indeed, the results presented in Figure 1 show that cell survival is independent of c-NHEJ function, which is equivalent to a partial collapse of this pathway in repair proficient cells” is simply wrong. The results in Figure 1 shows that survival depends on c-NHEJ function and the enhancement caused by the deficiency of this function is independent of the type of irradiation, or in other words, RBE of protons is independent of c-NHEJ function. Calculation and reporting of enhancement ratio produced by repair deficiency would clarify this issue.”

We thank the Reviewer for finding this error. We modified the section to avoid the identified problem.

“2.          The statement (page 10 lines 378-380) “The experimental testing of this hypothesis is confounded by the fact that while c- NHEJ entirely fails to contribute to cell survival after exposure to high-LET radiation modalities and RBE value of 1.1 is measured, it has a marked contribution when repair of all DSBs is assessed” is ambiguous since it is not clear what cells are meant – wild type or NHEJ deficient mutant. For wild type cells, NHEJ fails to contribute to cell survival at high LET resulting in high RBE values (~3 but not 1.1); for NHEJ deficient mutant, NHEJ cannot fail at high LET since it has already failed by definition (at any LET) resulting in low RBE (~1).”

Again, the Reviewer is right and we provide now the explanations requested.

“3.          Lines 77-78 page 2: What is meant by “high-LET quasi lesions”? Should it be “quasi high-LET lesions”?”

Indeed, we agree with the reviewer and used the suggested expression.

“4.          Lines 110 and 112 page 3: the energy of X-rays is measured in keV but not in kV. The operating voltage of X-ray tube is measured in kV.”

We thank the reviewer for pointing out this obvious mistake. We now use the correct units when appropriate.

“5.          Figure 4 legend – AA8 cells (wild type) label placed in Figures С and D and the reference to AA8 cells made in the text, however “HR-deficient cells” are indicated in the first sentence in the legend.”

The label of the Figure was corrected accordingly.